# A Dyadic Perspective of Felt Security: Does Partners’ Security Buffer the Effects of Actors’ Insecurity on Daily Commitment?

**DOI:** 10.3390/ijerph17207411

**Published:** 2020-10-12

**Authors:** Eri Sasaki, Nickola Overall

**Affiliations:** School of Psychology, University of Auckland, Auckland 1142, New Zealand; n.overall@auckland.ac.nz

**Keywords:** felt security, attachment anxiety, commitment, dyadic, strong link

## Abstract

Interdependence and attachment models have identified felt security as a critical foundation for commitment by orientating individuals towards relationship-promotion rather than self-protection. However, partners’ security also signals the relative safety to commit to relationships. The current investigation adopted a dyadic perspective to examine whether partners’ security acts as a strong link by buffering the negative effects of actors’ insecurity on daily commitment. Across two daily diary studies (Study 1, N = 78 dyads and Study 2, N = 73 dyads), actors’ X partners’ daily felt security interactions revealed a strong-link pattern: lower actors’ felt security on a given day predicted lower daily commitment, but these reductions were mitigated when partners reported higher levels of felt security that day. Actors’ X partners’ trait insecurity (attachment anxiety) interaction also showed this strong-link pattern in Study 1 but not Study 2. The results suggest that partners’ felt security can help individuals experiencing insecurity overcome their self-protective impulses and feel safe enough to commit to their relationship on a daily basis.

## 1. Introduction

Committing to relationships is accompanied by substantial risks—the more individuals invest in their relationship, the more vulnerable they are to the pain that will arise if partners are rejecting [1]. To fully commit, therefore, it is necessary to feel secure in a partner’s regard, availability and responsiveness [2,3,4]. During threatening interactions, for example, state feelings of security promote continued investment whereas feelings of insecurity prompt self-protective distancing [5]. Similarly, trait feelings of insecurity associated with attachment anxiety interferes with commitment [6,7].

However, partners’ security should also be an important signal of whether it is safe to commit to relationships. Partners’ insecurity heightens the risk of hurt and rejection that likely undermines commitment [4] whereas partners’ security signals the potential for a promising stable relationship that may promote commitment (e.g., [8,9]). Moreover, based on increasing evidence that partners play an important role in bolstering security [10], partners’ relative security may mitigate the effects of actors’ insecurity. In the current research, we adopt a dyadic perspective to examine whether actors’ and partners’ state and trait feelings of (in)security combine to shape commitment during daily interactions. In particular, we test whether partners’ security acts as a strong link by buffering the negative effect of actors’ insecurity on daily commitment.

### 1.1. The Need for Felt Security to Commit to Relationships

Commitment involves the motivation to maintain and persist in a relationship [11,12,13]. Commitment is the central factor determining whether people work to sustain relationship bonds [14], and thus, whether relationships remain intact or dissolve [15,16]. However, high levels of interdependence in romantic relationships produces two conflicting motives that make it challenging to remain committed on a daily basis: the need to remain committed and invested and the need to protect oneself against the risk of rejection [1,17]. To fully commit, individuals need to feel secure enough in their partners’ availability and responsiveness in order to be willing to open themselves up to the risk of hurt and rejection that accompanies commitment [4]. People who experience lower felt security, however, are more likely to protect themselves from expected rejection rather than prioritize sustaining their close relationships [18]. 

The proposition that security is a necessary condition for commitment is supported by research showing that state feelings of security determine whether people are inclined towards relationship-promotion versus self-protection during daily interactions. Lower felt security, involving doubts about the partner’s love, caring and responsiveness, leads to greater attempts to protect the self rather than invest in the relationship to minimize the possibility of hurt and rejection (see [4]). For example, lower felt security is associated with greater distancing and hostility following days of conflict and negative partner behavior [5]. By contrast, greater felt security, including feeling partners can be trusted to be loving and responsive, predicts attempts to counteract days of conflict and negative partner behavior by reaffirming partners’ acceptance and increasing closeness [5]. Thus, low versus high felt security disrupts versus promotes a range of commitment-related processes on a daily basis.

The importance of felt security in facilitating relationship promotion over self-protection is reflected in central tenets of attachment theory. Attachment theory [19] stipulates that the set goal of the attachment system is to attain felt security by promoting behavioral, cognitive and affective responses that garner protection and support from attachment figures (also [20]). Repeated or prolonged experiences of state felt security when interacting with caregivers creates individual differences in attachment functioning that determines trait levels of felt security in adult relationships [18]. In particular, based on a history of caregivers responding inconsistently to bids for love and support [21], attachment anxiety involves a preoccupation with attaining felt security in romantic relationships fueled by persistent fears of rejection and doubts about partners’ continued responsiveness [18]. For this reason, Fraley and Shaver [22,23] have argued that attachment anxiety is a sensitive indicator of people’s general (or trait) levels of felt security and safety in their relationship. In contrast to attachment anxiety, which reflects a hyper vigilant system involving persistent monitoring of feelings of security, Fraley and Shaver [22,23] outline that attachment avoidance reflects a motivational orientation that captures people’s down-regulation of felt security (see also [24]). Accordingly, people high in attachment avoidance persistently limit emotional closeness to minimize dependence rather than monitor security to maximize closeness and dependence [18]. We thus focus on attachment anxiety given that anxiety, not avoidance, captures the appraisal and experiences of felt security that is key to our theoretical framework.

The trait feelings of insecurity captured by attachment anxiety often interfere with people’s willingness and ability to commit to relationships [6,7]. For example, individuals higher in attachment anxiety report lower levels of commitment [9], and tend to be in shorter relationships that are more likely to end [25,26]. A recent large-scale effort using machine learning across 43 dyadic datasets confirmed the negative association between attachment anxiety and commitment [27]. That said, individuals higher in attachment anxiety also crave closeness and acceptance [18], and yearn for commitment in their relationships [28], which may explain why they are also more likely to commit to and remain in even unfulfilling relationships [6,29,30]. The conflict between heightened feelings of insecurity and potent desires for secure bonds may be why some studies have found nonsignificant and inconsistent links between attachment anxiety and global levels of commitment [25,31]. Another reason may be that partners’ levels of security also play an important role in the effects of individuals’ state and trait feelings of insecurity on commitment. 

### 1.2. A Dyadic Perspective: The Role of Partners’ Felt Security in Determining Commitment

Relationships, and whether they are worth investing in, involve the characteristics, feelings, thoughts, and behaviors of two partners. Accordingly, whether individuals (*actors*) are willing to commit to their relationship is also likely to be determined by their *partners*’ security. Partners who feel less secure likely generate relationship dynamics that heighten the risk of hurt and rejection and undermine actors’ daily commitment. Insecure partners are preoccupied with self-focused concerns and often put self-protection ahead of relationship promotion [4,18], resulting in them being less attentive and sensitive to actors’ needs and emotions (e.g., [32,33]), more clingy and needy (e.g., [34,35,36]), more critical and hostile during relationship-threatening events (e.g., [5,37,38,39]), less able to contain and recover from distressing situations [40], and thereby increasing the likelihood of daily difficulties [41]. Accordingly, partners’ insecurity is associated with actors perceiving insecure partners as needy, selfish, and unappreciative [5,42], and may signal a range of risks that reduces actors’ willingness to commit on a daily basis. Indeed, individuals with partners who are less secure report lower relationship commitment [8,9,27,43,44].

The importance of partners’ security in affecting commitment-related processes, and consequently signalling the relative safety to commit, highlights the dyadic nature of security. In particular, insecurity by either the actor or the partner may be a *weak link* and undermine commitment, even if the other is secure. Prior work examining dyadic patterns of commitment has demonstrated this type of weak-link pattern: one partner’s low commitment is enough to increase the risk of relationship dysfunction or dissolution even if the other is committed [45,46,47]. Likewise, Senchak and Leonard [48] found that couples in which one partner was high in attachment insecurity reported less intimacy-promoting behavior and felt less close as compared to couples in which both partners were secure. These dyadic weak-link patterns suggest that either actors’ or partners’ low felt security or attachment insecurity is enough to disrupt relationships because both signal the potential for partner rejection and hurtful relationship dynamics, and thus indicate that it is less safe to commit to relationships. Accordingly, it is possible that *both* actors and partners may need to experience felt security in order to sustain daily commitment, and commitment is eroded by either actors or partners feeling insecure.

However, there are good reasons to think that partners’ security could play a more positive, enhancing *strong-link* role by mitigating the negative effects of actors’ insecurity. In particular, a growing body of work has supported that partners can buffer the negative effects of insecurity by either activating mental representations of partners being available and responsive or through partners’ clear displays of love, support, and commitment (see [10,49,50]). For example, repeatedly priming security by either instructing participants to recall experiences of a close other being loving, sensitive and responsive, or exposing participants to the names of such a person, produces declines in attachment anxiety [51,52] and facilitates more responsive and supportive behavior towards a partner in need [53]. Partners’ actual loving and supportive behaviors can also enhance feelings of security in individuals high in attachment anxiety, such as when highly committed partners behave in more accommodative ways during conflict [54], exaggerate their expressions of affection [55], engage in physical touch [56,57], are responsive to individuals’ needs [58], or are secure enough to validate and promote individuals’ goals outside the relationships [59] (also see [49]). 

Based on these prior partner buffering effects, we propose that partners’ security will signal safety and a reason to trust in the strength of the relationship, and thus yield a strong-link pattern by mitigating the undermining effect of actors’ insecurity on daily commitment. Indeed, secure partners are more likely to evidence all of the buffering behaviors above as well as more generally activate a representation of an available and responsive partner. Secure partners are more willing to invest and maintain relationships [4,18], be more expressive, authentic, and affectionate [60,61,62,63], more responsive (e.g., [64,65]), more constructive during relationship-threatening events (e.g., [5,66,67,68]), are more likely to recover from distressing situations [69], and produce less conflict on a daily basis [41]. All of these benefits of partners’ felt security should promote a sense of safety that reduces the perceived risk of dependence and facilitates commitment even when actors are feeling insecure. Thus, we tested whether partners’ security would act as a *strong link* involving partners’ feelings of greater security mitigating the negative effects of actors’ low security on commitment.

### 1.3. Summary and Current Research

Interdependence and attachment models identify felt security as a critical foundation for commitment by orienting individuals towards investing in and promoting the relationship versus protecting the self from the vulnerability that dependence entails. However, both actors’ and partners’ felt security (or insecurity) should be important in minimizing (or highlighting) the risk of rejection and increasing (or reducing) commitment. In the current research, we recognize the dyadic nature of security by testing two research questions regarding how (RQ1) actors’ and partners’ state feelings of security and (RQ2) actors’ and partners’ trait insecurity combine to predict commitment during couples’ daily interactions. Although it is possible that partners’ insecurity signals a greater risk of rejection and thus acts as a weak link undermining even secure actors’ high daily commitment, based on recent research showing that partners can attenuate the damage of insecurity, we proposed that partners’ security would signal a sense of safety and thus act as a strong link by buffering the low daily commitment arising from actors’ insecurity.

In two dyadic daily sampling studies, both members of relatively committed couples reported their felt security and commitment at the end of each day over a 3-week period. It is important to examine felt security and commitment on a daily basis given that felt security can occur momentarily and show fluctuations (e.g., [51,52,53,59,70]), and commitment can also fluctuate at the daily, weekly, monthly, and yearly level [71,72,73]. Moreover, the links between felt security and commitment should be especially evident in the daily grind of people’s relationships where partners encounter a range of conflicting preferences, interests, and goals [1]. Thus, our primary research question (RQ1) focused on *state* felt security as actors’ and partners’ felt security in the moment determines how they negotiate the daily tension between investing in the relationship versus protecting themselves from the risk of rejection [4,5]. The relative sense of safety arising from actors’ and partners’ levels of daily felt security should shape daily commitment. In particular, in multilevel models assessing actor X partner state felt security on commitment during daily interactions, we tested whether actors’ low felt security would predict lower daily commitment, but partners’ high felt security would buffer the negative effects of actors’ low felt security.

Our second research question (RQ2) focused on the effects of *trait* levels of felt insecurity, as indexed by attachment anxiety, measured prior to the daily sampling procedure. Unlike state feelings of security that fluctuate day-to-day, attachment anxiety represents relatively stable, trait insecurities entailing intense desires for love accompanied by doubts about the partners’ dependability, creating vigilant monitoring of felt security [18,22,23]. We examined whether the expected effects of daily felt security also emerged for trait insecurity. In models testing the effects of actor X partner attachment anxiety on daily levels of commitment, we examined whether greater attachment anxiety would predict lower daily commitment, but this negative association would be buffered by partners’ lower attachment anxiety.

## 2. Materials and Methods 

Studies 1 and 2 consisted of two independent studies collected at different universities in different cities. All participants provided informed consent before they participated in each study. Each study was conducted in accordance with the Declaration of Helsinki, and the protocol was approved by the appropriate Ethics Committee at the University of Auckland (2010/527; Study 1) and Victoria University of Wellington (19287; Study 2). Given the procedures and analytic strategies were identical, and measures were very similar, we present the methods and results jointly for ease of comparison.

### 2.1. Participants

Both samples were recruited by advertisements posted across large city-based universities and associated organization (e.g., health and recreation centers) inviting couples who were involved in serious relationships (dating, cohabiting or married) of at least 1 year in length to participate in a study on daily relationship experiences. The target for each sample was 80 couples accounting for attrition due to non-compliance with the daily sampling procedure, which balanced funding with the aim to have adequate power to detect meaningful actor and partner effects based on prior studies and conventions at the time of data collection. The final sample for Study 1 involved 78 heterosexual couples who were in serious dating relationships (48.7%), married (10.3%) or cohabitating (34.6%), with the remainder in casual or steady relationships. Relationship length ranged from 0.5 to 10.25 years (*M* = 2.58, *SD* = 1.99), and couples’ ages ranged from 17 to 48 years (*M* = 22.44, *SD* = 4.82). The final sample for Study 2 involved 73 heterosexual couples who were in serious dating relationships (41.1%), married (13.7%) or cohabiting (35.6%), with the remainder in steady relationships. Relationship length ranged from 0.42 to 21 years (*M* = 3.19, *SD* = 3.53), and couples’ ages ranged from 18 to 48 years (*M* = 23.61, *SD* = 6.86). Additional analyses presented in the Online Appendix A (OSM) showed that longer relationship length was associated with lower attachment anxiety in Study 1 but not in Study 2, and relationship length was not associated with daily felt security and commitment. There were also no differences across relationship length in the effects of actors’ X partners’ felt security and actors’ X partners’ attachment anxiety.

The original aims of the study when funded did not focus on actor X partner effects, and thus a priori power analyses were not conducted. See OSM for power considerations and further details regarding these samples.

### 2.2. Materials and Procedure

During an initial session, couples completed questionnaires assessing attachment anxiety. Participants then received detailed instructions for completing a 3-week daily diary. Scales were constructed by averaging items. Table 1 presents descriptive statistics.

*Attachment security*. In both studies, participants completed the Adult Attachment Questionnaire [39], which is a widely used scale that involves two subscales assessing two attachment dimensions. Nine items assessed attachment anxiety (e.g., “I often worry that my romantic partners don’t really love me”), and eight items assessed avoidance (e.g., “I’m not very comfortable having to depend on romantic partners”; 1= *strongly disagree*, 7 = *strongly agree*).

#### Daily diary

At the end of each day for 21 consecutive days, participants completed a web-based record reporting their felt security and commitment. On average, participants completed 19.3 and 19.1 diary records producing 3276 and 2786 daily observations for Study 1 and Study 2, respectively.

*Daily felt security*. In Study 1, participants rated three items assessing felt security that day (“I felt insecure about our relationship” [reverse-coded], “I felt confident that my partner loves me”, and “I trusted that my partner would be there if I needed him/her”; 1 = *not at all*, 7 = *very much*). In Study 2, participants rated three similar items (“I felt like my relationship was strong and secure”, “I felt confident that my partner would continue to love me in the future”, and “I trusted that my partner would be there if I needed him/her”; 1 = *not at all*, 7 = *very much*).

*Daily commitment*. Participants rated a single item in Study 1 (“today, I was committed to our relationship”; 1 = *not at all*, 7 = *very much*) and in Study 2 (“how committed were you to your relationship today?”; 1 = *not at all*, 7 = *extremely*).

## 3. Results

In both studies, the relatively weak correlations between (in)security across partners indicated that couple members could hold different levels of security. The average correlation between actor and partner felt security within each daily assessment was 0.41 (range from 0.27 to 0.60) and 0.31 (range from 0.05 to 0.56) for Study 1 and Study 2 respectively. The zero-order correlations between actor and partner attachment anxiety were even smaller (−0.06 in Study 1 and 0.17 in Study 2). 

We report two sets of analyses to test the two research questions. We first tested our primary research question (RQ1) which focused on assessing the degree to which the expected negative effect of actors’ lower daily felt security on daily commitment was buffered by partners’ greater daily felt security. We then examined our secondary research question (RQ2) exploring whether the expected negative effects of trait insecurity (greater attachment anxiety) was buffered by partners’ trait security (lower attachment anxiety). We tested the two research questions separately given that they represent distinct processes, and then ran additional analyses to examine the independence of any specific dyadic effects that emerged. 

### 3.1. RQ1: Dyadic Effects of Daily Felt Security on Commitment

We conducted separate analyses for Studies 1 and 2. In all analyses, we controlled for the main effect of the distinguishing variable gender (−1 = women, 1 = men; see [74]). Additional tests revealed no gender differences in the effects of actors’ X partners’ felt security and actors’ X partners’ attachment anxiety (see OSM). 

To test whether actors’ and partners’ daily felt security interacted to predict daily commitment, we followed the procedures outlined by Kenny et al. [74] to analyze repeated measures dyadic data (see OSM for annotated syntax). We modelled the degree to which within-person variations in (a) actors’ felt security, (b) partners’ felt security, and the (c) interaction between actors’ and partners’ felt security predicted actors’ commitment on the same day, controlling for actors’ commitment the previous day to ensure that any effects that emerged did not stem from lingering effects of the previous day [75]. Predictor variables were person-centered. To ensure that the effects assessed daily variations in actors’ and partners’ felt security, we also controlled for the between-person main and interaction effects of actors’ and partners’ felt security [75]. The central aims focus on the daily (within-person) effects of felt-security. The between-person effects are presented in the OSM. In both studies, higher average levels of partners’ felt security across days enhanced the higher commitment experienced by actors who had higher average felt security across days. 

The results focusing on the within-person effects for each study are shown in Table 2.

In both studies, lower actors’ felt security was associated with lower daily commitment. In Study 2, but not Study 1, lower partners’ felt security was also associated with lower commitment. Moreover, in both studies, the actor X partner felt security interaction was significant. Decomposing the interaction revealed a strong-link pattern (see Figure 1): lower actors’ felt security was associated with lower daily commitment, but these drops were greater on days when partners also felt less secure (Study 1: *b* = 0.48, *t* = 23.73, *p* < 0.001, *r* = 0.56; Study 2: *b* =0.42, *t* = 20.10, *p* < 0.001, *r* = 0.51), compared to days when partners reported greater felt security (Study 1: *b* = 0.37, *t* = 13.74, *p* < 0.001, *r* = 0.36; Study 2: *b* = 0.26, *t* = 10.27, *p* < 0.001, *r* = 0.29). Accordingly, the lower commitment experienced by actors on days they reported lower felt security (left side of figure) was significantly attenuated on days partners felt more secure (Study 1: *b* = 0.07, *t* = 3.32, *p* = 0.001, *r* = 0.09; Study 2: *b* = 0.18, *t* = 8.56, *p* < 0.001, *r* = 0.25 in Study 2).

### 3.2. RQ2: Dyadic Effects of Trait Insecurity on Commitment

Next, we examined whether actors’ X partners’ attachment anxiety predicted daily commitment. We conducted separate models for Studies 1 and 2. Using the dyadic regression approach outlined by Kenny et al. [74], we modelled the degree to which (a) actors’ attachment anxiety, (b) partners’ attachment anxiety, and the (c) interaction between actors’ and partners’ attachment anxiety predicted actors’ daily levels of commitment (see OSM for annotated syntax). The results are shown in Table 3.

In both studies, actors’ lower security (greater attachment anxiety) predicted lower commitment across daily life. In Study 1, but not Study 2, partners’ lower security (greater attachment anxiety) also predicted lower daily commitment. Moreover, in Study 1, but not Study 2, the actor X partner attachment anxiety interaction was significant. Decomposing the interaction revealed a strong-link pattern (see Figure 2a): greater actors’ attachment anxiety was associated with lower daily commitment when partners were also higher in attachment anxiety (*b* = −0.37, *t* = −3.63, *p* < 0.001, *r* = 0.39), but not when partners were lower in attachment anxiety (*b* = 0.03, *t* = 0.43, *p* = 0.668, *r* = 0.05). Accordingly, the lower commitment experienced by actors high in attachment anxiety (right side Figure 2a) was eliminated when partners were low in attachment anxiety (*b* = −0.45, *t* = −4.49, *p* < 0.001, *r* = 0.46).

### 3.3. Integrative Data Analyses

Given that Studies 1 and 2 involved similar procedures and measures, it is possible to conduct integrative data analysis (IDA) by pooling the data across both studies to maximize power and to test whether the effects were replicated across the two studies (see [77,78]). Using the pooled dataset, we reran the same analyses for RQ1 (as presented in Table 2) and RQ2 (as presented in Table 3) and included the main and interaction effects of sample membership (−1 = Study 1, 1 = Study 2). Full results are shown in the OSM. IDA revealed that there were study differences in the effects of both (1) actor X partner state felt security and (2) actor X partner trait insecurity. For RQ1, the actor x partner state felt security interaction was significant in both studies but stronger in Study 2 than Study 1 (see Figure 1). The differences in the strength of the dyadic daily effect could have arisen due to slightly different measures of felt security, other differences in the two samples (see Method Section), or other factors responsible for variation in the daily effects. For RQ2, the actor x partner trait insecurity was only significant in Study 1 and not in Study 2 (as shown in Figure 2). These differences indicated that interpreting the pooled effects from IDA is not justified and showed that the effects for RQ2 were not replicated across studies.

### 3.4. Additional Analyses 

The analyses across both Study 1 and 2 provided support that partners’ daily felt security buffered the negative within-person effects of actors’ low felt security on daily commitment. This strong-link pattern only emerged for trait insecurity (attachment anxiety) in Study 1, and was not replicated in Study 2. Given that greater attachment anxiety predicted lower daily levels of felt security (*b* = −0.24, *t* = −4.50, *p* < 0.001, *r* = 0.37 in Study 1 and *b* = −0.10, *t* = −1.99, *p* = 0.049, *r* = 0.17 in Study 2), we also ran additional analyses to show that the dyadic effects of felt security in the moment was independent of trait levels of insecurity. Simultaneously modelling the main and interaction effects of both (1) actors’ and partners’ daily felt security (as in Table 2) and (2) actors’ and partners attachment anxiety (as in Table 3) on daily commitment produced identical results as when felt security and attachment anxiety were modelled separately (see OSM for full results).

Finally, we focused on attachment anxiety given that attachment anxiety captures the tendency to appraise, experience and monitor felt security [22,23]. Attachment avoidance, by contrast, reflects a motivational orientation that continually down-regulates closeness and dependence (rather than reflexively monitors felt security to obtain closeness and dependence). Consistent with this persistent defensiveness, rerunning the same models as in Table 3 with attachment avoidance instead of anxiety revealed that greater actors’ attachment avoidance predicted lower daily commitment, but no partner or actor X partner interaction effects emerged (see OSM). Simultaneously modeling the main and interaction effects of (1) actors’ and partners’ attachment anxiety and (2) actors’ and partners’ attachment avoidance also did not alter the effects of attachment anxiety shown in Table 3 (see OSM).

## 4. Discussion

The current investigation adopted a dyadic perspective to test whether partners’ security buffers the negative effects of actors’ insecurity on daily commitment. Consistent with both attachment and interdependence frameworks that emphasize felt security is a critical foundation for commitment by facilitating relationship-promotion rather than self-protection, both daily feelings of insecurity and trait insecurity (attachment anxiety) predicted lower levels of commitment during couples’ daily lives. However, illustrating the important role that partners’ security plays in signaling safety and enhancing commitment in relationships, actor X partner daily felt security interactions revealed a strong-link buffering effect of partners’ security that replicated across two daily sampling studies: actors’ daily feelings of insecurity predicted within-person reductions in commitment, but these reductions were mitigated when partners reported high levels of felt security that day. When testing the actor X partner trait insecurity (attachment anxiety) interaction, this strong-link pattern only occurred in Study 1 and not Study 2. In sum, three of four tests provided support that partners’ security plays an important role in helping people who experience insecurity feel safe enough to commit to their relationship on a daily basis. 

### The Strong-Link Buffering Effect of Partners’ Security: Contributions, Implications and Future Directions

Prior research has provided evidence that both actors’ and partners’ (in)security signals the relative safety (or risk) to commit to a relationship [4,6,8]. Our dyadic approach advances prior demonstrations of actor and partner effects by providing preliminary evidence that actors’ and partners’ felt security jointly combine to shape daily commitment. Consistent with a range of evidence that partners can buffer actors’ insecurity [10,49,50], the results indicate that partners’ felt security acts as a strong link by attenuating the negative effect of actors’ felt insecurity on daily commitment. Thus, partners’ felt security appears to be an important signal of whether it is safe to commit to relationships, particularly when people’s own feelings of insecurity are making it hard to overcome the impulse to self-protect in order to prioritize and invest in their relationship. 

There are many ways in which partners’ felt security should signal a sense of safety and reduce the risk of dependence to help insecure people commit to their relationships. More secure partners exhibit a range of pro-relationship cognitive and behavioral processes in daily interactions that indicate that they are more willing and able to assuage actors’ feelings of insecurity. In general, partners who feel more secure are more committed, more available and responsive (e.g., [64,65]), more constructive during relationship-threatening events (e.g., [5,66,67]), and more likely to recover from and be less affected by negative relationship events [41,69]. These pro-relationship responses capture many partner behaviors involving communicating commitment, affection and support that have been shown to buffer insecurity in prior research (e.g., [54,55,56,57,59]). 

Unlike prior research focusing on specific partner buffering behaviors during couples’ interactions, however, the current studies identify that partners’ own reported feelings of state security can reduce the negative effects of felt insecurity on commitment. On the one hand, this is a considerable strength given that partners’ security is likely an important precursor to the types of partner behaviors that alleviate the destructive responses arising from individuals’ insecurity. On the other hand, the current studies did not identify the ways partners’ security is conveyed or perceived in daily interactions to provide the safety needed to override drops in commitment when individuals feel insecure. It is likely that the range of cognitive and behavioral processes associated with partners’ felt security are implicated, although which specific process is at play on a given day probably depends on the idiosyncratic demands and needs of the couple that day. This new evidence for the dyadic effects of felt security in couples’ daily life indicates that identifying how partners’ felt security is communicated and interpreted within couple interactions is an important, novel direction for future research. 

Examining felt security within couples’ daily interactions provides valuable insights into attachment processes occurring in the moment. During daily life, couples often have to negotiate different personalities and conflicting interests that can activate the attachment system. Understanding how momentary fluctuations in felt security influence actors’ and partners’ immediate responses informs how the attachment system functions within these daily interactions. It may also be the case that attachment processes occurring in the moment differ from more general patterns of reactions arising from trait security. In the current studies, the strong-link dyadic effects were more robust for state feelings of security as compared to trait security (low attachment anxiety). Limited power might have contributed to the inconsistent dyadic effects of trait insecurity compared to the more powerful design of repeated assessments of state felt security. It is also possible that differences in measurement may have accounted for the state versus trait differences in the current research. The strong-link effects might be driven more strongly by the presence of security, which was captured by our assessments of felt security each day, rather than the absence of insecurity, as reflected by our assessment of low attachment anxiety. This potentially important distinction indicates that adult attachment theory and research should pay more careful attention to the presence of attachment security, rather than just low levels of attachment insecurity (anxiety or avoidance), to help advance understanding of how to enhance state and trait security. 

Even if the strong-link buffering processes examined are strongest when assessing state rather than trait (in)security, daily attachment and felt-security processes should nonetheless feed outward to shape relationship outcomes across time. Our results illustrate that, within a particular day, interactions with partners who feel more secure buffer the negative effects of actors’ state feelings of insecurity on daily commitment. Repeated daily interactions and buffering experiences of partners’ security should help promote more general pro-relationship outcomes across days as well as trait security in couples’ relationships [51,52], especially given the positive daily outcomes the strong-link effects have, such as protecting commitment. Thus, partner buffering of daily feelings of insecurity may lay the groundwork to promote more committed and stable relationships and enhance trait attachment security across time [49]. Exploring the temporal profiles in the dyadic patterns of felt security provides an interesting avenue for future research.

We focused on daily commitment because it is a critical indicator of whether people feel safe in their dependence to invest in their relationship rather than self-protect on a daily basis. The strong-link pattern in which partners’ security protected daily commitment from actors’ insecurity has important implications for the future of the relationship. Commitment is a powerful indicator that relationships will persist [79,80] because commitment motivates various behaviors and responses that sustain relationships, including sustaining more charitable views of partners (e.g., [81]), derogating attractive alternatives (e.g., [82]), enacting more constructive and less destructive behaviors (e.g., [83]), sacrificing when encountering conflicting goals [84], and forgiving partners’ transgressions (e.g., [85]). Moreover, daily drops in commitment have important effects even for couples high in commitment or trait security as indicated by fluctuations in commitment predicting poorer relationship outcomes [71]. By protecting against daily fluctuations in commitment, partners’ daily security may not only protect against the risk of long-term declines in relationship quality but also facilitate more pro-relationship responses by actors in ways that reinforce both partners’ security and commitment each day, creating a cyclical process of mutual growth that promotes relationships across time [86]. Examining the ways in which strong-link dyadic patterns during couples’ daily life could generate positive longitudinal outcomes for both partners is a valuable direction for future research.

Our dyadic perspective examining the interaction effects of actors’ and partners’ felt security on daily commitment provides insight into naturally-occurring attachment processes during couples’ daily interactions, but is also accompanied by the limitations of correlational data preventing causal conclusions about the direction of effects across partners. For example, the dyadic effect can be interpreted as partners’ security buffering the detrimental effects of actors’ insecurity as theorized. Alternatively, actors’ insecurity could also interfere with the benefits of partners’ security. Our theoretical account is more aligned with existing work demonstrating evidence that partners play an important role in bolstering security (see [10,49,50]). Nevertheless, both accounts fit with a dyadic perspective emphasizing the interdependent nature of attachment and commitment processes [18,87]. Individuals’ outcomes are not just determined by their own or their partners’ levels of felt security and reactions, but by the joint effect of both their own and their partners’ feelings and behavior. Thus, we believe that future work devoted to identifying how to buffer insecurity or enhance security would benefit from understanding how both actors’ and partners’ (in)security create dyadic patterns that could protect, promote, or impede relationships. 

Conceptualizing felt security as a dyadic process provides new targets for couples’ intervention. Insecurity is often conceptualized as an individual process and vulnerability, which may create a focus on the dyad member who has trait insecurity. Yet, it is the difficult moments in relationships that couples need to successfully navigate to sustain commitment and satisfaction, and it is the relative feelings of (in)security in those moments that create the destructive self-protective responses that are most often targeted during therapy [88]. Rather than focusing on specific behavioral patterns, the current results highlight the importance of both partners’ relative felt security, and in particular, emphasize that partners’ felt security could be leveraged as a source of safety. Treating both partners’ security as equally important may be advantageous by helping both partners attend to feelings of security (not just insecurity) in order to recognize the potential for either partner to be a strong link when this matters. It might also help expand a focus on specific buffering strategies to consider the ways that security is idiosyncratically expressed, and can be facilitated, within a given couple [49]. This approach offers a balanced involvement of both dyad members helping each other to manage felt security during important relationship interactions. The partner feeling less secure could depend on the other’s security, while the partner who feels more secure could take on the role of a strong link, creating safety by expressing security. 

In sum, guided by interdependence and attachment models, the current studies represent the first tests of dyadic patterns of felt security in daily life. However, despite offering promising initial findings that partners’ felt security plays a strong-link buffering role in daily life, the current studies only provide preliminary evidence. Our discussion of the implications of the current results acknowledged limitations of our investigation, including small samples sizes that may have limited power for testing dyadic patterns, differences in the measures and strength of effects across the two studies, correlational data preventing causal conclusions about the direction of effects across actors’ and partners’ (in)security, and open questions regarding the various mechanisms through which partners’ security is conveyed and perceived. Nevertheless, taken together, we believe the studies offer promising initial evidence that suggests that future investigations will benefit from adopting a dyadic perspective to advance understanding of the effects of felt security and the ways that insecurity might be buffered in daily relationship interactions.

## 5. Conclusions

The current studies adopted a dyadic perspective to consider whether partners’ felt security may enhance relationships by buffering the negative effects of actors’ felt insecurity on daily commitment. The results suggest that partners’ felt security acts as a strong link by protecting actors’ daily commitment from the often damaging effects of actors’ daily feelings of insecurity. In two studies assessing the dyadic dynamics of security across the course of couples’ daily lives, actors’ low feelings of security on a given day predicted lower daily levels of commitment. However, partners’ felt security mitigated this negative effect, thereby protecting relationships from the damage arising from fluctuations in commitment. In one of the studies, this strong-link pattern was also evident when assessing trait levels of insecurity (attachment anxiety). The results provide promising evidence that partners’ security can help individuals who feel insecure overcome their self-protective impulses and feel safe enough to commit to their relationship on a daily basis. 

## Figures and Tables

**Figure 1 ijerph-17-07411-f001:**
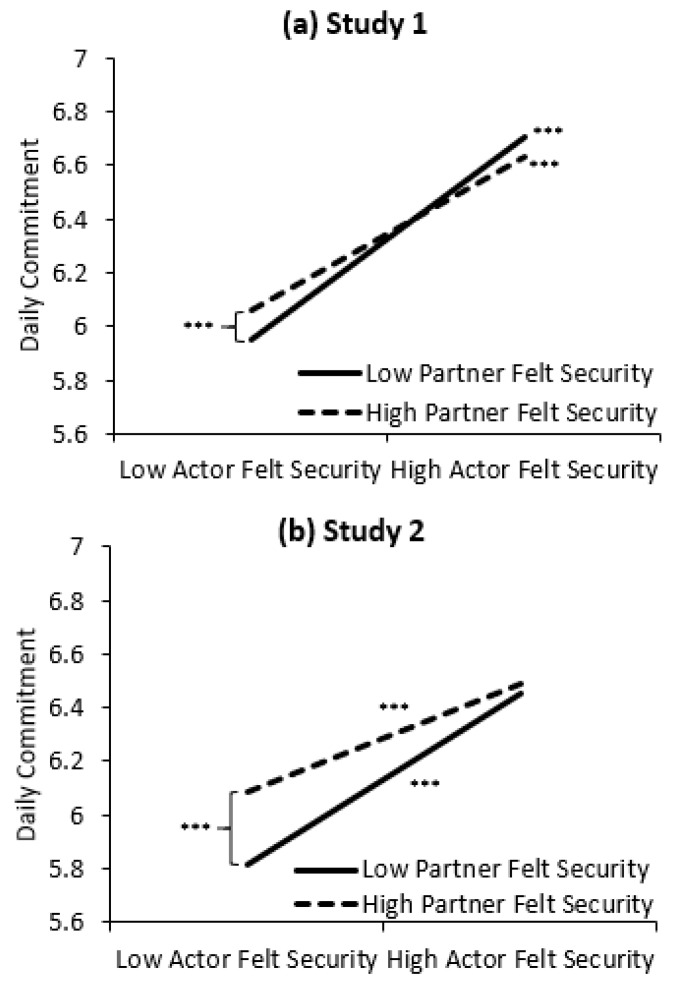
The effects of actor and partner felt security on daily commitment in (**a**) Study 1 and (**b**) Study 2. *Note*. Low and high levels of actor and partner felt security represent 1 SD below and above the mean. The simple effects of the slopes and contrasts are marked *** *p* ≤ 0.001.

**Figure 2 ijerph-17-07411-f002:**
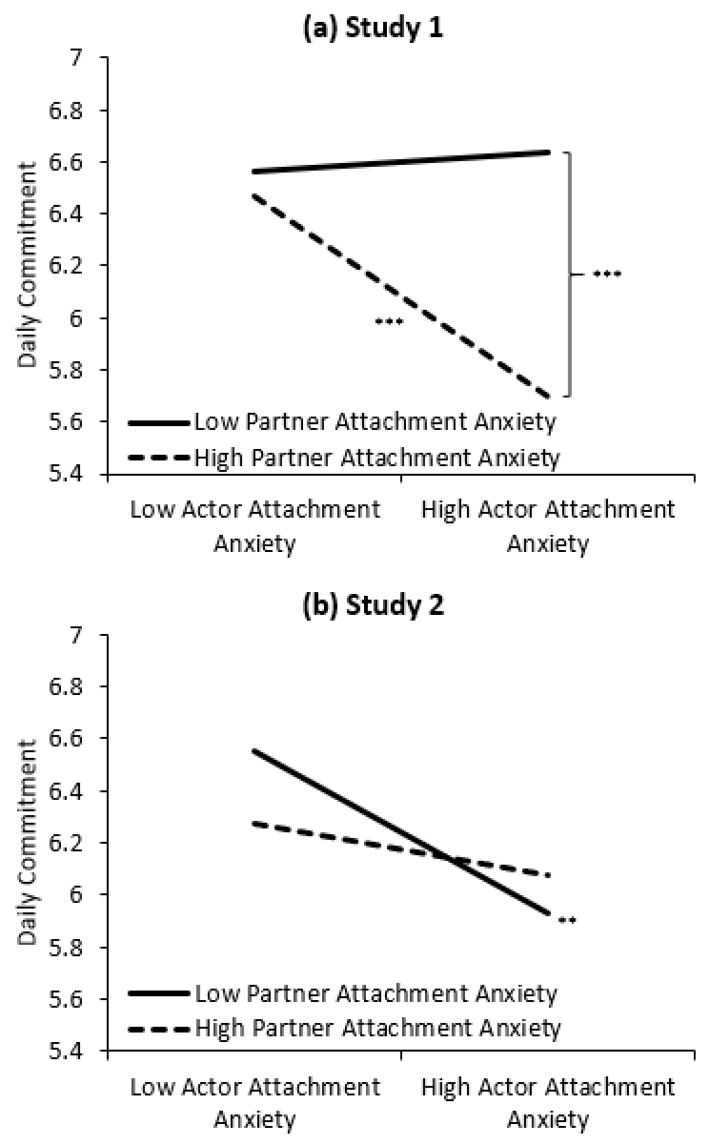
The effects of actor and partner attachment anxiety on daily commitment in (**a**) Study 1 and (**b**) Study 2. *Note*. The interaction effect was not significant in Study 2 but was presented for comparison across studies. Low and high levels of actor and partner attachment anxiety represent 1 SD below and above the mean. The simple effects of the slopes and contrasts are marked *** *p* < 0.001 and ** *p* < 0.01.

**Table 1 ijerph-17-07411-t001:** Descriptive statistics of all measures.

Measures	Study 1			Study 2		
	*Mean*	*(SD)*	*R*	*Mean*	*(SD)*	*R*
Questionnaire measures						
Attachment Anxiety	2.99	(1.05)	0.80	3.10	(1.09)	0.84
Attachment Avoidance	2.92	(1.04)	0.77	2.90	(0.92)	0.72
Daily measures						
Felt Security	6.20	(1.12)	0.95	6.27	(1.03)	0.96
Commitment	6.37	(1.15)	-	6.23	(1.15)	-

All measures represent averages across items on 1 to 7 Likert-type scales. Daily measures represent averages of daily assessments across the 21-day diary period. R = reliability. Reliability for questionnaire measures uses Cronbach’s alpha (α) to assess the internal consistency of the scale items, while the reliability for daily felt security (*R*_C_) refers to the reliability of within-person change. No reliability is given for commitment as commitment was assessed with a single item each day.

**Table 2 ijerph-17-07411-t002:** The effects of actor and partner daily felt security on actors’ daily commitment.

	Commitment					
			*95% CI*			
Predictors	*B*	*t*	Low	High	*p*	*r*
**Study 1**						
Intercept	6.34	115.47	6.228	6.447	<0.001	1.00
Actor Felt Security	0.42	21.50	0.382	0.459	<0.001	0.41
Partner Felt Security	0.01	0.56	−0.027	0.049	0.574	0.01
Actor × Partner Felt Security	**−0.07**	**−4.21**	**−0.102**	**−0.037**	**<0.001**	**0.12**
**Study 2**						
Intercept	6.21	168.11	6.137	6.284	<0.001	1.00
Actor Felt Security	0.34	18.74	0.302	0.373	<0.001	0.39
Partner Felt Security	0.10	5.61	0.066	0.137	<0.001	0.13
Actor × Partner Felt Security	**−0.10**	**−5.30**	**−0.138**	**−0.064**	**<0.001**	**0.16**

Analyses were conducted controlling for the corresponding between-person effects of felt security. The significant interaction effects presented in bold are presented in Figure 1. CI = confidence interval. Effect sizes (*r*) were computed using Rosenthal and Rosnow’s [76] formula: *r* = √(*t 2* / *t 2* + *df*). In these multilevel models, the Satterthwaite approximation is applied to provide specific degrees of freedom for each effect, which were used to calculate the effect sizes.

**Table 3 ijerph-17-07411-t003:** The effects of actor and partner attachment anxiety on actors’ daily commitment.

	Commitment					
			*95% CI*			
Predictors	*B*	*t*	Low	High	*p*	*r*
**Study 1**						
Intercept	6.34	91.86	6.205	6.480	<0.001	1.00
Actor Attachment Anxiety	−0.17	−2.90	−0.281	−0.053	0.004	0.25
Partner Attachment Anxiety	−0.25	−4.29	−0.362	−0.133	<0.001	0.36
Actor × Partner Attachment Anxiety	**−0.19**	**−2.84**	**−0.328**	**−0.057**	**0.006**	**0.31**
**Study 2**						
Intercept	6.21	73.86	6.039	6.374	<0.001	0.99
Actor Attachment Anxiety	−0.19	−2.86	−0.320	−0.058	0.005	0.24
Partner Attachment Anxiety	−0.03	−0.45	−0.161	0.102	0.655	0.04
Actor × Partner Attachment Anxiety	**0.09**	**1.27**	**−0.052**	**0.234**	**0.208**	**0.15**

The interaction effects presented in bold are presented in Figure 2. CI = confidence interval. Effect sizes (*r*) were computed using Rosenthal and Rosnow’s [76] formula: *r* = √(*t 2*/*t 2* + *df*). In these multilevel models, the Satterthwaite approximation is applied to provide specific degrees of freedom for each effect, which were used to calculate the effect sizes.

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
