# Peer review of "A Dyadic Perspective of Felt Security: Does Partners’ Security Buffer the Effects of Actors’ Insecurity on Daily Commitment?"

_ijerph, 2020, doi:10.3390/ijerph17207411_

Round 1

Reviewer 1 Report

The current study examined the buffering effects of a partner’s felt security on the link between actor felt security and commitment in two samples of couples who completed daily diaries. The authors found support for what they proposed—actor felt insecurity was not as predictive of commitment when their partner reported high felt security. The effects were more inconsistent when examining chronic attachment anxiety. The paper was well-written and everything was mostly straight-forward. Some of the analytic choices were a little puzzling, so I think it’s worth doing a few additional analyses to strengthen the inferences that are being made. First, I think an overall model that includes both attachment anxiety and avoidance (and their cross-partner interactions) and the felt security (and the cross-partner interactions) should be run. Running piecemeal analyses switching out one variable for another in the supplement seemed a bit strange when I think most people would most care about a model that did that (not just controlling for avoidance and having all of these variables included). I also thought it was strange that, from what I could tell from the syntax, what wasn’t done was a parsing of between and within person effects like is often done with diary data (e.g., entering between-person felt security and a within-person felt security that is a deviation from that between person average; see Bolger & Laurenceau, 2013). Otherwise, the study seems to conflate these types of effects. Thus, it’s hard to know if the felt security is a between-subjects effect (e.g., I’m paired with a partner who feels security most of the time v. not) or a within-subjects effect (e.g., my partner felt particularly secure today (relative to their average) so it buffers the aforementioned negative effect). I think doing the analysis this way is at least worthwhile to improve inferences. On page 4 (ln 157-158): there’s an odd statement: “…assessing within-person variation in actor x partner state felt security on…” Integrating a partner effect wouldn’t technically be “within-person” but I see what you mean (e.g., a component of a repeated measures). Might be good to reword it though. How were sample sizes determined? I think that should be added to the manuscript. Or at least some discussion of power—what were the number of observations? What’s the smallest effect that could be found given this? Throughout the manuscript, I also had the inkling that the interpretation could be considered from the other perspective as well. In other words, instead of partners buffering against insecurity: a paper could also be easily written in which actor insecurity interferes with the positive influence of partner security (e.g., focusing on the individual’s volition instead of their partner). Is there a more circumspect way that the authors could talk about this in the Discussion? Or at least acknowledge that, from what I can tell, this would be a statistically identical/valid way of talking about it? Interestingly, I probably would’ve just started from the beginning as an integrative data analysis with sample as a covariate/moderator. I think this analysis was a good thing to do (and I’d probably even suggest it for the felt security analyses given that the individual samples were so small, and noting the large caveats). Although the samples/measures were largely comparable, do the authors have any data for how highly correlated the felt security and commitment variables are with each other? Also, in text, I think most people would care the most about the pooled effect of the interaction (not just whether the interaction differed across sample), but it’s never reported. They should report that because it’s not significant as a main test which effectively doubled the sample. In the Discussion, the authors say that they didn’t identify the ways partner security is conveyed or manifested as a limitation (i.e., what’s the exact mechanism behind these effects and what occurs in the actual context of the relationship?). I would maybe amend the sentence to also include how their partner interprets/filters these displays as well—it’s not just how they’re conveyed and expressed. There might be some bias for how they’re perceived. On the top of page 11, the authors talk about “partners who are consistently successful at buffering in-the-moment feelings…” This was a little confusing because it merges some between-subject language (partners who…) and within-subject language (are consistently successful—always higher relative to their b/w person average). Maybe the aforementioned analyses could help with this, but maybe it could be worded a little less definitively b/c that might not have been perfectly tested (even in this type of diary study—they just reported on daily things, not in-the-moment event-based stuff). On the bottom of page 11, the authors talk about how they think there’s a “dance” of exchanges that happen between partners. I think this goes a bit beyond the data and could be toned down a bit (or just removed).

Reviewer 2 Report

The manuscript is undoubtedly interesting, and the presented research brings us closer to a better understanding of the determinants of commitment to a relationship and building a lasting, satisfying relationship. The authors attempted to define the importance of partners' felt security for the actor's commitment. In the conducted research, they decided to check whether the partner's higher felt of security may weaken the negative effects of the actor's low  security on his commitment. The studies took into account both chronic feelings of (in) security and daily felt security.

I have formulated the following comments which I believe would help to enhance the manuscript.

  • It would be worth clarifying what is meant by commitment in the context of daily commitment and determine how it is related to commitment treated as a component of love for example in Sternberg's triangular theory of love or an element/characteristic of long-term relationships.
  • It would be good to indicate what is the justification which allows us to assume that the belief that you are loved by your partner and that he supports you, when we need it, may change from day to day / may be subject to significant fluctuations as well as determine the relationship between daily felt security and chronic feelings of security.
  • I propose to articulate the questions and hypotheses more clearly.
  • In my opinion, it would be good to characterize the group of respondents in more detail. It is not clear what the criterion of "serious relationships" was. The criteria for selecting people for the research were not given. It is not known what the minimum length of relationship was. It is not known if all the couples who agreed to participate filled out the diaries for 21 days. From the data presented, it can be concluded that the respondents were in the period of emerging adulthood, and most of the couples had a relatively short experience, which would be worth taking into account when formulating conclusions. Given that there are grounds to believe that security may be related to length of relationship, which may be related to commitment.
  • I propose to complete the description of the Adult Attachment Questionnaire, indicate in what form the results are obtained and how they are interpreted. There are no psychometric characteristics of the questionnaire in the text.
  • It is not known why in Study 1 and Study 2 slightly different daily diary items were proposed. One may also wonder whether the statement "today, I was committed to our relationship" is a measure of commitment or a subjective assessment by the actor. Then the question arises whether the partner's daily felt security is related to the actor's daily commitment or to his assessment of commitment. If you want to determine the level of commitment in a relationship, it might be worth distinguishing indicators of this involvement and on their basis, assess whether it was high or low. It might be interesting to include the assessment of commitment in the relationship made by the actor (by himself) and by his/her partner.
  • When considering the relationship between the partner's daily felt security and the actor's daily commitment, maybe it is worth to think about the partner's daily commitments as an intermediary variable. There is a possibility that the partner's high level of felt security leads to his high commitment to the relationship, which in turn affects the actor's commitment.
  • It would be good to point out the limitations of the research.

Round 2

Reviewer 1 Report

Thanks to the authors for being flexible. I don’t fully agree with some of the points they made (e.g., they acknowledge they didn’t have [and didn’t mean to insinuate they had] power to test the interaction but proceeded to do it). But many of the disagreements come down to some arguable and philosophical differences and I don’t want to be adversarial/obstructionist and “rail road” a paper over these issues. I hate reviewers that do that, so I don’t want to be that person! But I did have two suggestions:

1) That’s great that between and within effects were partitioned and the syntax looks great. What’s missing is the actual results of the between subjects effects. A table in the supplement where you report both would be great, rather than just saying it was controlled for. Meta-analysts might want that information and it’d be good to know if the stable between-subjects effects matter. They don’t need to be mentioned or talked about at length, just made available to readers in some way.

2) For point number 7 in your response, I apologize for not being clearer. My issue was that the studies used different measures of felt security. When I was asking about the correlations, I was asking how highly correlated the items/construct in Study 1 (that contains “I felt confident that my partner loves me.”) were with the items/construct in Study 2 (that contains “I trusted that my partner would be there if I needed him/her.”). I was asking if you did any diagnostic study/additional data collection to see if these things were indeed the same construct or if it was uncritically accepted as being the same thing. My guess is that this additional validation work probably wasn’t done. Although the actor effects in Table 2 were similar (as the authors note in their reply), the partner effects were dramatically different. So it might be worth at least acknowledging via a footnote or in the Discussion that the Studies didn’t use identical instruments. The studies differed a bit in their Results which could be the result of many things. But one obvious one is that stuff was measured differently. Again, the two felt security measures are probably correlated highly and “seem similar” but acknowledging that they were different as a limitation could be worthwhile.
